# Immune Checkpoints Inhibitors and Chemotherapy as First-Line Treatment for Metastatic Urothelial Carcinoma: A Network Meta-Analysis of Randomized Phase III Clinical Trials

**DOI:** 10.3390/cancers13061484

**Published:** 2021-03-23

**Authors:** Hsiao-Ling Chen, Vinson Wai-Shun Chan, Yu-Kang Tu, Erica On-Ting Chan, Hsiu-Mei Chang, Yung-Shun Juan, Jeremy Yuen-Chun Teoh, Hsiang Ying Lee

**Affiliations:** 1Department of Pharmacy, Kaohsiung Municipal Ta-Tung Hospital, Kaohsiung 801, Taiwan; 1058065@kmuh.org.tw (H.-L.C.); 880504@kmhk.org.tw (H.-M.C.); 2School of Medicine, Faculty of Medicine and Health, University of Leeds, Leeds LS2 9LU, UK; vinson.chan@doctors.org.uk; 3Institute of Epidemiology and Preventive Medicine, National Taiwan University, Taipei 100, Taiwan; yukangtu@ntu.edu.tw; 4Department of Medical Research, National Taiwan University Hospital, Taipei 100, Taiwan; 5S.H. Ho Urology Centre, Department of Surgery, Prince of Wales Hospital, The Chinese University of Hong Kong, Hong Kong, China; ericachan@surgery.cuhk.edu.hk; 6Department of Urology, Kaohsiung Municipal Ta-Tung Hospital, Kaohsiung 801, Taiwan; 840066@kmuh.org.tw; 7Graduate Institute of Clinical Medicine, College of Medicine, Kaohsiung Medical University, Kaohsiung 807, Taiwan; 8Department of Urology, Kaohsiung Medical University Hospital, Kaohsiung 807, Taiwan; 9Department of Urology, School of Medicine, College of Medicine, Kaohsiung Medical University, Kaohsiung 807, Taiwan

**Keywords:** immune checkpoints inhibitors, urothelial carcinoma, chemotherapy, network meta-analysis

## Abstract

**Simple Summary:**

On the basis of the efficacy and tolerable safety profiles of immune checkpoints inhibitors (ICIs) in second-line metastatic urothelial carcinoma (mUC) patients, some emerging clinical trials focus on the first-line treatment. Thus, we conducted a network meta-analysis (NMA) to assess and compare the response and toxicity of ICIs in naïve-chemotherapy mUC setting. According to our results, combination therapy (either ICIs plus chemotherapy (CTX) or ICIs plus ICIs) had a higher priority in terms of overall survival. Concerning monotherapy, ICIs are not inferior to CTX in terms of OS. In view of the adverse effect, ICIs are very tolerable, and combination therapy did not lead to a higher incidence of grade 3–5 AEs when compared with CTX.

**Abstract:**

Immune checkpoints inhibitors (ICIs) were considered as second-line treatments in metastatic urothelial carcinoma (mUC) based on better survival benefit and safety profile than chemotherapy (CTX). We aimed to assess different ICIs regimens in the efficacy and safety for front-line treatments in mUC patients. A comprehensive literature search was performed and Phase II-III randomized controlled trials (RCTs) on ICIs for patients with mUC were included. The outcome was evaluated by overall survival (OS), progression of free survival (PFS), objective response rate (ORR), and grade 3–5 adverse events. Network meta-analysis was used to estimate the effect size. Surface under cumulative ranking curves (SUCRAs) were applied to rank the included treatments for each outcome. Results: The survival benefit of a single ICI was non-inferiority to chemotherapy (CTX). Although no superior effects were indicated, combination therapy (either ICIs plus CTX or ICIs plus ICIs) presented better OS compared with CTX alone. In terms of PFS, combination therapy produced a noticeable benefit over CTX. Regarding the SUCRA ranking, atezolizumab plus CTX was associated with the best ranking for OS and pembrolizumab plus CTX was the best in PFS. In terms of safety, a single ICI had better safety profile than CTX and combination therapy had a similar risk of grade 3–5 adverse events with CTX. Conclusions: Our NMA results revealed that combination therapy has better ranking compared with monotherapy in OS and acceptable AEs. ICIs alone present non-inferior OS but a lower incidence of AEs compared with CTX.

## 1. Introduction

Metastatic urothelial carcinoma (mUC) arising from upper or lower urinary tract systems has poor prognosis and short-term survival [1]. Currently, platinum (cisplatin or carboplatin) based chemotherapy is considered the first-line treatment for mUC with an overall response rate (ORR) of 40–60% and median overall survival (OS) of 14–15 months [2,3,4]. However, there are some challenges when patients are unfit or intolerable to receive chemotherapy. The most commonly used platinum-based regimens are methotrexate, vinblastine, doxorubicin, and cisplatin (MVAC), as well as gemcitabine plus cisplatin (GC), but they may not be suitable in patients with multiple comorbidities, poor performance status, and renal function impairment [5]. Therefore, alternative first-line treatment replacing chemotherapy for mUC is an important issue in clinical practice. 

In recent years, immune checkpoints inhibitors (ICIs) against programmed death 1 (PD-1) and its ligand (PD-L1) have been investigated for mUC which progress despite chemotherapy. Five immune checkpoints inhibitors including two PD-1 inhibitors, pembrolizumab and nivolumab and three PD-L1 inhibitors, as well as atezolizumab, avelumab, and durvalumab have been approved by the Food and Drug Administration for mUC patients with prior chemotherapy or cisplatin ineligible condition [6]. Although UC is a chemo-sensitive cancer, a significant proportion of patients still experience disease progression after first-line chemotherapy. There are several agents that have been investigated in the second-line setting including vinflunine, taxanes, checkpoints inhibitors, anti-angiogenic therapy, etc., [7,8]. A previous network meta-analysis evaluated various second-line therapeutic agents and concluded that there is no evidence to support that one agent is better than the other in terms of OS. However, PD-1/L1 inhibitors can be considered as they are associated with lower toxicity profile [9]. 

With the promising results of PD-1/L1 inhibitors as a second-line treatment for mUC, there are several emerging clinical trials focusing on PD-1/L1 inhibitors for mUC in the first-line setting. However, no head to head RCTs compare the efficacy and safety among ICIs and combination therapy (either ICIs plus chemotherapy (CTX) or ICIs plus ICIs). As ICIs are very expensive and mUC patients have short-term survival, we urgently need to identify the single best available treatment. Hence, we conducted a network meta-analysis (NMA) to investigate and compare the treatment response and toxicity of PD-1/L1 inhibitors versus chemotherapy as a first-line treatment for patients with mUC. 

## 2. Results

### 2.1. Literature Search

The search strategy was presented in Appendix A and a total of 1388 studies were imported by the comprehensive search strategy. After eliminating duplication, 1328 studies underwent title/abstract screening and 1294 of them were excluded due to irrelevance, resulting in 34 studies for a full text review. At the end of the process, 13 published studies retrieved from three completed RCTs met our inclusion criteria and remained for qualitative synthesis. The PRISMA flow diagram is presented in Figure 1. 

### 2.2. Study Characteristics and Quality Evaluation

Details on the characteristics of three included studies were reported in Table 1. All of them were conducted for patients with newly diagnosed locally advanced or metastatic UC without previous therapy. A parallel three-arm design was predominated to compare the effects between chemotherapy alone, ICIs plus chemotherapy, and a combination of different ICIs. PD-1 inhibitor was examined in one RCT (pembrolizumab was used in KEYNOTE-361), PD-L1 inhibitors were examined in two RCTS (atezolizumab was used in IMvigor130, durvalumab was used in DANUBE), and tremelimumab (a CTLA4 inhibitor was combined with durvalumab in DANUBE). The median age was 67–69 and the percentage of males across the trials was ranged from 72 to 80%. In terms of metastatic site, 17~24% subjects were reported with lymph node metastasis and 74~80% subjects were reported with visceral metastases, including lung, liver, and bone.

Results of the quality assessment were presented in Appendix B (Figure A1). High risk was assessed in the performance bias since the open label design was conducted in KEYNOTE-361, IMvigor130, and DANUBE. Due to the fact that limited data about KEYNOTE-361 were provided from the annual meeting of 2020 ESMO, quality was unclear in sequence generation, allocation concealment, and outcome assessment.

### 2.3. Efficacy and Safety Evaluation from the Network Meta-Analysis

Figure 2 presented the network constructions of eligible comparisons. For OS, ORR, and grade 3–5 adverse events (Figure 2a), seven interventions were included in the network analysis, such as chemotherapy (abbreviation to CTX), ICI alone (atezolizumab, embrolizumab, and durvalumab, abbreviation to Atezo, Pembro, and Durva), ICIs plus CTX (Atezo plus CTX and Pembro plus CTX), and a combination of different ICIs (durvalumab plus tremelimumab, abbreviation to Durva plus Treme). For PFS (Figure 2b), five interventions were included (CTX alone, combination therapy, and durvalumab monotherapy). The effect size of the pairwise comparisons, SUCRA rankings, and probability to be the best treatment were presented in Figure 3 and Figure 4 and Appendix C. Additionally, the subgroup analysis in OS by gender, age, and cisplatin eligibility were summarized in league tables and presented in Appendix D.

#### 2.3.1. Overall Survival

Results were presented in Figure 3a. Although no superior effects were indicated, combination therapy presented lower HRs compared with CTX alone (HR = 0.83, 95% CI = 0.69–1.00 for Atezo plus CTX, HR = 0.86, 95% CI = 0.72–1.03 for Pembro plus CTX, HR = 0.85, 95% CI = 0.72–1.00 for Durva plus Treme). In terms of monotherapy, the survival benefit of single ICI was non-inferiority to CTX alone (HR = 1.02, 95% CI = 0.83–1.25 for Atezo, HR = 0.92, 95% CI = 0.77–1.10 for Pembro, HR = 0.99, 95% CI = 0.83–1.18 for Durva). In addition, no significant differences were presented among combination therapy (HR = 1.04, 95% CI = 0.80–1.34 for Pembro plus CTX vs. Atezo plus CTX, HR = 1.02, 95% CI = 0.80–1.31 for Durva plus Treme vs. Atezo plus CTX, HR = 0.99, 95% CI = 0.77–1.26 for Durva plus Treme vs. Pembro plus CTX) and similar effects were presented among monotherapy (HR = 0.93, 95% CI = 0.72–1.19 for Pembro vs. Durva, HR = 1.03, 95% CI = 0.79–1.35 for Atezo vs. Durva, HR = 1.11, 95% CI = 0.84–1.46 for Atezo vs. Pembro). Regarding the SUCRA ranking (Figure 4a), the probability of Atezo plus CTX was associated with the best ranking for OS (highest SUCRA and Prbest value, SUCRA = 80.2%, Prbest = 38.6%, Figure A2), followed by Durva plus Treme (SUCRA = 75.6%), Pembro plus CTX (SUCRA = 70.6%), Pembro (SUCRA = 50.8%), Durva (SUCRA = 29.7%), CTX alone (SUCRA = 22.0%,) and Atezo (SUCRA = 21.7%).

#### 2.3.2. Progression Free Survival

Results were presented in Figure 3b. ICIs plus CTX produced a noticeable benefit over CTX alone (HR = 0.82, 95% CI = 0.70–0.96 for Atezo plus CTX, HR = 0.78, 95% CI = 0.65–0.94 for Pembro plus CTX) but a similar effect was presented between different regimens of ICIs plus CTX (HR = 0.95, 95% CI = 0.75–1.21 for Pembro plus CTX vs. Atezo plus CTX). Durva was associated with a worse performance in PFS compared with CTX alone (HR = 1.16, 95% CI = 1.10–1.22 for Durva plus Treme, HR = 1.28, 95% CI = 1.03–1.59 for Durva). Additionally, the probability of Pembro plus CTX had the highest SUCRA and Prbest scores (SUCRA = 90.8%, Prbest = 63.4%, Figure 4b and Figure A2), followed by Atezo plus CTX (SUCRA = 84.0%), CTX (SUCRA = 49.0%), Durva plus Treme and (SUCRA = 20.0%), and Durva (SUCRA = 5.2%).

#### 2.3.3. Overall Response Rate

Results were presented in Figure 3c. In terms of combination therapy, Pembro plus CTX provided a greater ORR than CTX (response ratio = 1.22, 95% CI = 1.05–1.42), Durva plus Treme provided the worse ORR than CTX (response ratio = 0.68, 95% CI = 0.57–0.80), and no significant difference was detected between Atezo plus CTX and CTX (response ratio = 1.05, 95% CI = 0.90–1.22). Accordingly, Durva plus Treme had the worse ORR than Pembro plus CTX (response ratio = 0.55, 95% CI = 0.44–0.70) and Atezo plus CTX (response ratio = 0.64, 95% CI = 0.51–0.81). In terms of monotherapy, ICI alone gave less improvement in ORR compared with CTX alone (response ratio = 0.52, 95% CI = 0.42–0.65 for Atezo, response ratio = 0.67, 95% CI = 0.55–0.83 for Pembro, response ratio = 0.48, 95% CI = 0.39–0.59 for Durva). Moreover, Pembro plus CTX was regarded to have a better objective response rate with the highest SUCRA and Prbest scores (SUCRA = 98.9%, Prbest = 93.5%, Figure 4c and Figure A2), followed by Atezo plus CTX (SUCRA = 79.7%), CTX (SUCRA = 71.4%), Pembro (SUCRA = 41.0%), Durva plus Treme and (SUCRA = 40.9%), Atezo (SUCRA = 13.2%), and Durva (SUCRA = 4.9%).

#### 2.3.4. Grade 3–5 Adverse Events

Results were presented in Figure 3d. We found no significant differences in the risk of grade 3–5 adverse events between ICIs plus CTX and CTX (risk ratio = 1.00, 95% CI = 0.96–1.05 for Atezo plus CTX, risk ratio = 1.07, 95% CI = 1.00–1.14 for Pembro plus CTX). Additionally, a lower risk was observed among Durva plus Treme users compared with CTX (risk ratio = 0.89, 95% CI = 0.81–0.97 for Durva plus Treme). In terms of monotherapy, patients with ICIs presented better safety profile than CTX alone (risk ratio = 0.55, 95% CI = 0.49–0.61 for Atezo, risk ratio = 0.77, 95% CI = 0.70–0.85 for Pembro and risk ratio = 0.75, 95% CI = 0.67–0.84 for Durva) and patients with Atezo presented a significantly lower risk than Pembro and Durva (risk ratio = 0.71 95% CI = 0.62–0.83 for Pembro, risk ratio = 0.73, 95% CI = 0.63–0.85 for Durva). Based on the SUCRA value, that larger SUCRA value indicated the lower risk of adverse events. Atezo had the best safety profile (SUCRA = 100%, Prbest = 100% Figure 4d and Figure A2), followed by Durva (SUCRA = 77.3%), Pembro (SUCRA = 72.2%), Durva plus Treme (SUCRA = 50.2%), CTX (SUCRA = 25.3%), Atezo + CTX (SUCRA = 23.9%), and Pembro + CTX (SUCRA = 1.0%).

#### 2.3.5. Subgroup Analysis

Results of the subgroup analysis were presented in Appendix D (Figure A3). Although lower HRs were observed among ICIs plus CTX, the HRs were close to 1 and no statistical differences were presented between ICIs plus CTX and CTX. However, Atezo + CTX users were associated with superior effects compared with CTX among patients more than 65 years old (HR = 0.80, 95% CI = 0.66–0.97) or patients that received cisplatin (HR = 0.73, 95% CI = 0.55–0.97). For patients younger than 65 years old, Durva plus Treme was regarded as the best regimen in OS with a significantly improved benefit compared with CTX (HR = 0.68, 95% CI = 0.52–0.91) and highest SUCRA (SUCRA = 85.9%).

#### 2.3.6. Consistency and Transitivity

Based on Table 1 and Figure 2, the network plot showed that each of the three loops is formed by a three-arm trial, and therefore, for any direct treatment comparison, its direct and indirect evidence comes from the same trial. Consequently, by definition, the evidence is always consistent.

Based on Appendix E (Table A1), the balanced distribution was presented at baseline among patients that received chemotherapy in DANUBE, KEYNOTE-361, and IMvigor130. Therefore, chemotherapy was allowed to be the common comparator for a valid network comparison.

## 3. Discussion

Although platinum-based chemotherapy is considered the first-line treatment for UC, it is inevitably associated with some limitations. For example, some patients are unfit to receive chemotherapy due to comorbidities, and some patients progress despite receiving chemotherapy. Recent clinical trials have established the efficacy of atezolizumab as a second-line treatment for locally advanced and mUC, which led to its FDA approval in 2016 [10,11]. KEYNOTE-045 Phase III trial showed that pembrolizumab conferred better OS over second-line chemotherapy for mUC [12]. On the basis of the promising results from two Phase II studies (cohort 1 of the IMvigor210 trial and KEYNOTE-052), FDA has also approved atezolizumab and pembrolizumab as a first-line treatment for cisplatin-ineligible mUC patients [13,14].

While the results of ICIs are very promising, comparisons among ICIs were currently unknown. Hence, we performed this network meta-analysis on the available Phase III trials (IMvigor130, DANUBE, KEYNOTE-361) to investigate the role of ICIs and combination therapy (either ICIs plus CTX or ICIs plus ICIs) in the first-line setting for both mUC cisplatin-ineligible and eligible patients [15,16,17]. SUCRA analysis is used to rank these included agents. In KEYNOTE-361 trial, it showed improvement in OS and PFS in pembrolizumab plus CTX compared with CTX alone, however, these results did not meet statistical significance. In DANUBE trial, durvalumab plus tremelimumab also have no significant improvement in OS even in a high PD-L1 population. In contrast, atezolizumab plus CTX presents a significant benefit in both OS (HR: 0.83; *p* = 0.027) and PFS (HR: 0.82; *p* = 0.007) from IMvigor130 trial.

In our analysis, the addition of pembrolizumab (or atezolizumab) to platinum-based chemotherapy resulted in better PFS and ORR when compared with CTX alone. Although there was no significant overall survival benefit, a better median overall survival was observed in the combination therapy. Chemotherapy has an immunosuppressive effect, but when combined with ICIs, it may impose an immunomodulatory function. By modulating the tumor microenvironment, the body’s immune response may be influenced resulting in a better efficacy upon ICIs [18]. ICIs are more effective when targeting tumors that have an inflamed phenotype, defined as the presence of type I interferon (IFN) activation, immune potentiating chemokines that attract T cells, antigen presentation, cytotoxic effector molecules, and CD8 + T cells. Theoretically, the addition of other ICIs or CTX may convert “non-inflamed” tumors into “inflamed” tumors, and the efficacy of ICIs may be improved [19].

In terms of monotherapy, although a lower response rate was noted upon ICIs, ICIs are not inferior to CTX in terms of OS. Hence, it is reasonable to give ICIs to mUC patients who cannot tolerate the adverse effects of CTX. Given the low response rate of ICIs, combined with another treatment that targets a mutated gene, for example, fibroblast growth factor receptor (FGFR) seems to be a reasonable approach [20]. More research is needed in this area, particularly on the profiling of gene signaling pathways. Based on a previous retrospective study, the benefits of ICIs in bladder cancer patients can be long-lasting and persistent over 10 months despite discontinuation. Therefore, for patients who are on the ICIs treatment, the treatment duration may be shortened if they experience any significant toxicities and/or any financial restrictions [21].

As expected, our analysis showed that ICIs were associated with a lower incidence of adverse events when compared with CTX. Additionally, no significant differences between ICIs plus CTX and CTX were presented. Adverse effects related to ICIs are termed “immune-related adverse effects (irAEs)” which may affect multiple organs simultaneously or at different time points. However, these irAEs are usually manageable [22]. Even if the patient has to discontinue ICIs due to irAEs, the treatment responses after discontinuation of ICIs treatment may still be long-lasting.

Upon further subgroup SUCRA analysis, combination therapy was shown to have better OS in both male or female, young people or elderly, cisplatin eligible or ineligible patients, primary tumor in upper or lower tract. Although ICIs might be better for elderly patients due to its favorable safety profile, an impaired immune system with increasing age may alleviate the efficacy of ICIs [23,24]. Therefore, combination therapy may still be considered a reasonable treatment strategy to enhance the efficacy of ICIs in well-selected patients. Gender differences have been proposed to contribute to the treatment response towards ICIs due to the diversities of innate and adaptive immunity between males and females [25]. However, our analysis showed that combination therapy could achieve a better OS in both males and females. In the subgroup of cisplatin ineligible patients, combination therapy also has a high priority in terms of OS. While we wish to maximize the synergistic effects of combination therapy, we must balance between its treatment efficacy and safety profile. Future prospective trials may focus on the dosage and the order of administration of these drugs [26].

There are several limitations in our study. First, only three Phase III RCTs were included for NMA and the available data is relatively limited. The baseline characteristics of the included patients may be heterogenous across the studies, and the results may be affected. Second, for a more comprehensive comparison, we assessed OS, PFS, ORR, and the severity of AEs in our analysis. However, not all survival rates were analyzed in RCTs and might have an impact on the evaluation of the best treatment modality. Furthermore, upper tract urothelial carcinoma (UTUC) patients comprising of renal pelvis and ureter cancer were also included in this analysis. The molecular characterization between UTUC and bladder cancer may differ and require different clinical management strategies [27]. Unfortunately, we were unable to analyze this based on the existing data.

## 4. Materials and Methods

This study followed the preferred reporting items for systematic reviews and meta-analyses (PRISMA) extension statement for the network meta-analysis [28].

### 4.1. Search Strategy and Study Selection

We identified eligible studies through a comprehensive search from Medline/PubMed, Embase, Cochrane library (CENTRAL and CDSR), and Clinical-Trials.gov up to 15 October 2020 without language limitation. Moreover, we searched the abstracts in the main oncology congresses database, such as the American Society of Clinical Oncology (ASCO), European Medical Oncology (ESMO), and American Association of Cancer Research (AACR). In addition, we reviewed the reference lists of the retrieved studies to include more relevant studies. The following search terms were used in the search strategy: Transitional cell carcinoma (TCC), urothelial carcinoma (UC), immunotherapy (atezolizumab, durvalumab, avelumab, pembrolizumab, nivolumab, tremelimumab, ipilimumab or ramucirumab), and chemotherapy for TCC or UC. The inclusion criteria were as follows: (1) Completed Phase II–III RCTs involving an adult with advanced or metastatic TCC or UC, (2) the RCTs focused on newly diagnosed treatment-naïve patients, (3) the RCTs provided the efficacy and safety comparison between immunotherapy and the standard treatment arm with chemotherapy.

### 4.2. Data Extraction and Quality Assessment

Two independent reviewers (H.Y. Li and Y.C. Teoh) performed literature screening, data extraction, and quality assessment. In the case of any discrepancies, a third reviewer (H.L. Chen) was introduced for discussion to reach a consensus. The extracted information included trial name, published year, trial phase, NCT number (a RCT identifier of ClinicalTrials.gov), baseline characters, treatment arm with a subject number, and median follow up duration. We evaluated the quality of included RCTs by the Cochrane Collaboration’s risk of bias (ROB) tool and presented the results by the Review Manager (RevMan, version 5.1, The Nordic Cochrane Centre: The Cochrane Collaboration, Copenhagen, Denmark) [29].

### 4.3. Data Synthesis and Statistical Analysis

Our primary outcome was the treatment efficacy, which was evaluated by overall survival (OS), progression of free survival (PFS), as well as the objective response rate (ORR). Our secondary outcome was the safety profile focusing on grade 3–5 adverse events according to the Common Terminology Criteria for Adverse Events (CTCAE) [30]. For binary indicators, the response ratio was used as the effect size for ORR and the risk ratio was used for adverse events. For time dependent indicators, such as OS and PFS, the adjusted hazard ratio (HR) was regarded as the effect size. If HR was not provided from the published trials, it was calculated from Kaplan Meier (K-M) survival curves based on the algorithm by Guyot et al. [31].

For the data synthesis, we first generated the network geometry in order to clarify which interventions were compared directly or indirectly. After that, the network meta-analysis (NMA) was applied. NMA is a technique for comparing interventions in a single analysis by combining direct and indirect evidence across the network of available studies. If two particular interventions have never been compared head-to-head, but these two have both been compared to a common comparator (placebo or standard therapy), then an indirect comparison can be evaluated vs. the common comparator. We undertook a network meta-analysis under the frequentist framework by the mvmeta Stata command (version 16, Stata, College Station, TX, USA) [32] to estimate the effect size for pairwise comparisons. The contrast based analysis was performed for multiple treatment comparisons with the restricted maximum likelihood approach. In addition, NMA allows ranking all treatment arms in the network geometry and identifying which is the best and worst among them based on the surface under cumulative ranking curve (SUCRA). We calculated the SUCRA and probability of being best (Prbest) to rank all the included treatments for each outcome. The larger SUCRA value stood for the better rank of the intervention effect and lower risk of adverse events.

Consistency and transitivity were checked for the validation of NMA. Inconsistency was defined as the difference between direct and indirect evidence. The design by treatment inconsistency model was used to check the consistency for each direct treatment contrast based on the difference between the different study designs. Transitivity was evaluated by comparing the distribution of effect modifiers across different comparisons.

## 5. Conclusions

From this NMA focusing on the first-line treatment of mUC patients, combination therapy (either ICIs plus CTX or ICIs plus ICIs) had a higher priority in terms of OS upon the ranking analysis. Concerning monotherapy, ICIs are not inferior to CTX in terms of OS. The benefit of combination therapy is presented regardless of gender and age upon the subgroup analysis. In view of the adverse effect, ICIs are very tolerable, and combination therapy did not lead to a higher incidence of grade 3–5 AEs when compared with CTX.

## Figures and Tables

**Figure 1 cancers-13-01484-f001:**
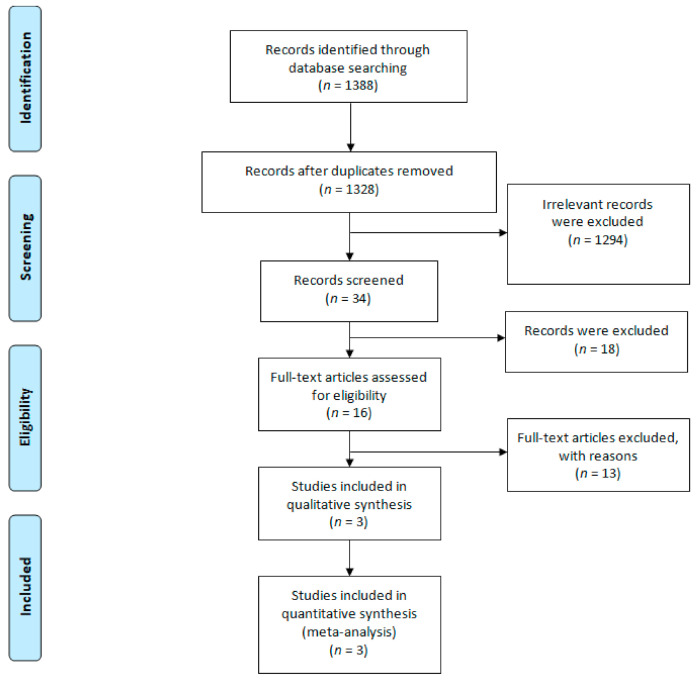
PRISMA flow diagram.

**Figure 2 cancers-13-01484-f002:**
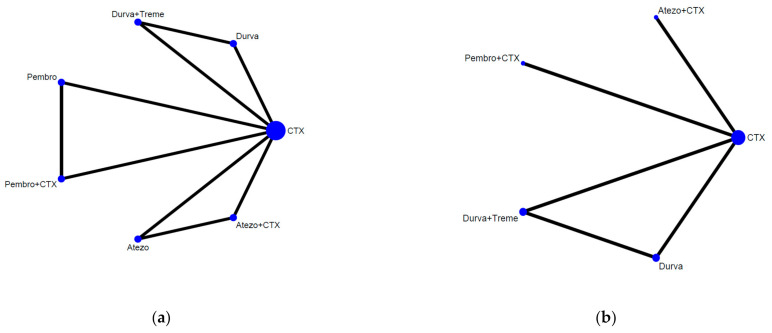
Network constructions for comparison in overall survival (OS), progression of free survival (PFS), objective response rate (ORR), and grade 3–5 adverse events (AEs). (**a**) Network constructions for comparison in OS (hazard ratio (HR)), ORR, grade 3-5 AEs. (**b**) Network constructions for comparison in PFS (HR).

**Figure 3 cancers-13-01484-f003:**
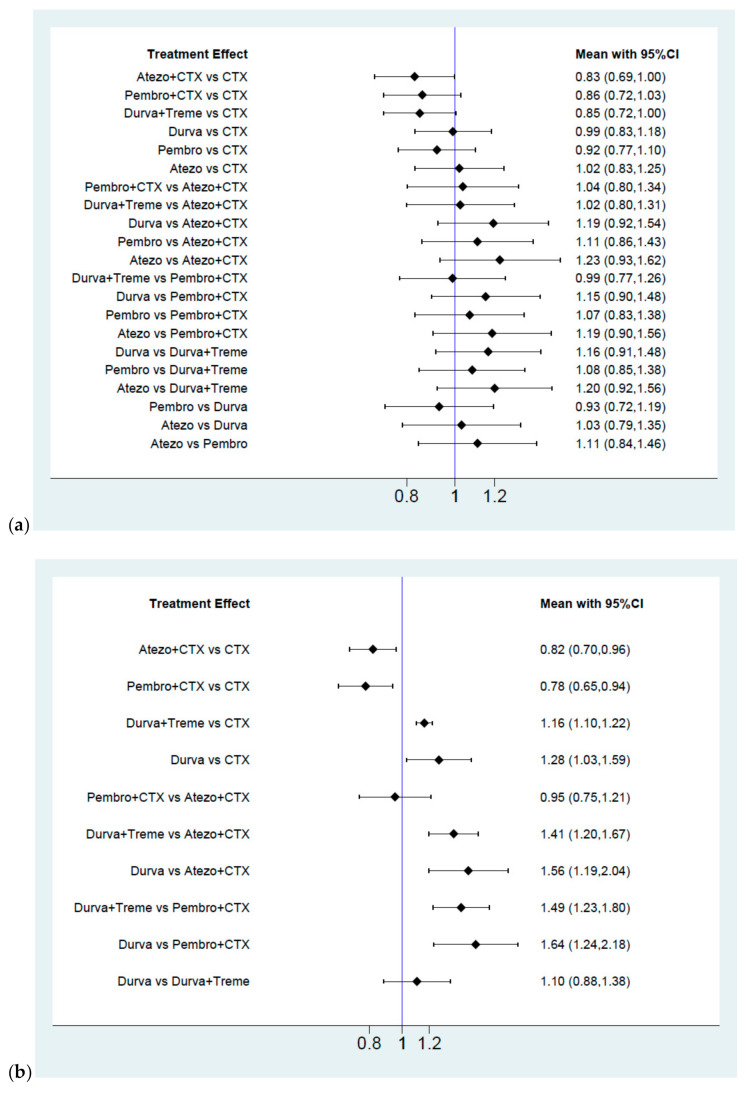
Summary of effect size for pairwise comparison. (**a**) Hazard ratio for overall survival, (**b**) hazard ratio for progression free survival, (**c**) response ratio for overall response rate, (**d**) risk ratio for grade 3–5 AEs.

**Figure 4 cancers-13-01484-f004:**
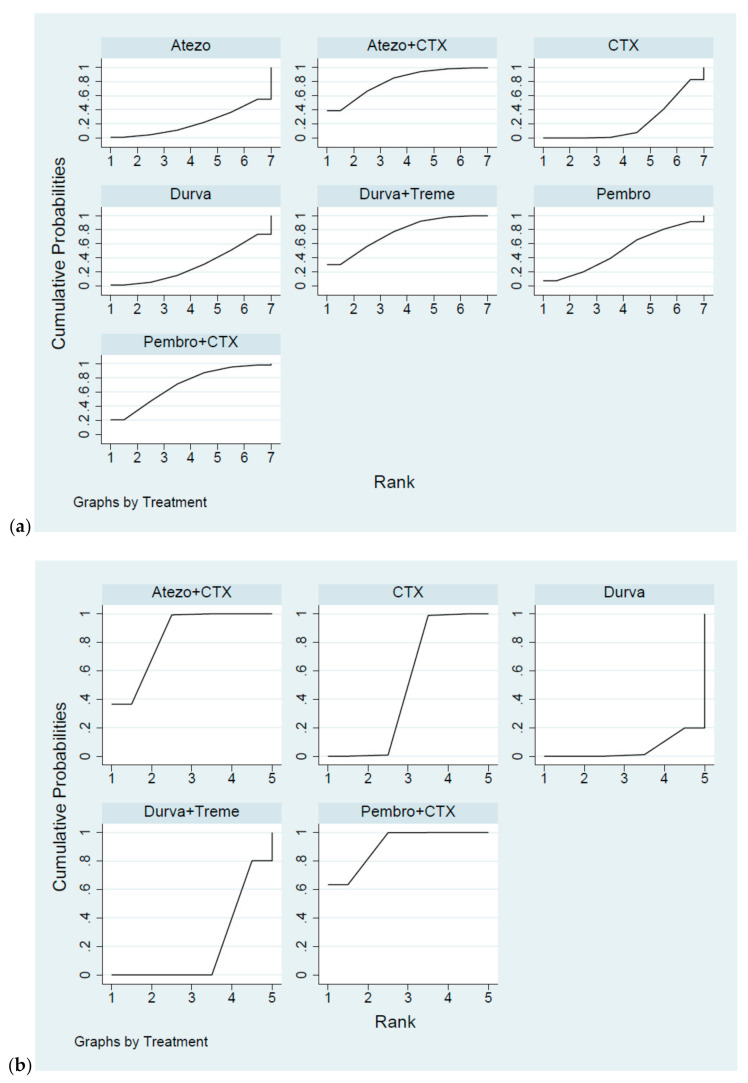
Cumulative ranking probability for different interventions. (**a**) Hazard ratio for overall survival, (**b**) hazard ratio for progression free survival, (**c**) response ratio for overall response rate, (**d**) risk ratio for grade 3–5 AEs.

**Table 1 cancers-13-01484-t001:** Characteristics of the included studies.

Trial Name	Year	NCT Number	Phase	ICI-Based Treatment	ICI Category	Design	Stage	Median Age	Males (%)	Site of Metastatic Disease (%)	Treatment Arm (Patient Number)
IMvigor130	2020	NCT02807636	3	atezolizumab	PD-L1 inhibitors	three arms	advanced	67–69	75–77	Lymph node only (17.89%)	1. atezolizumab (360)
						open-label	or metastatic			Visceral metastases (79.96%)	2. atezolizumab + platinum-based CTX (451)
											3. platinum-based CTX (400)
DANUBE	2020	NCT02516241	3	durvalumab	PD-L1 inhibitors	three arms	advanced	67–68	72–80	Lymph node only (20.45%)	1. durvalumab (346)
				tremelimumab	CTLA4 inhibitors	open-label	or metastatic			Visceral metastases (79.36%)	2. durvalumab + tremelimumab (342)
											3. platinum-based CTX (344)
KEYNOTE-361	2020	NCT02853305	3	pembrolizumab	PD-1 inhibitors	three arms	advanced	68~69	74.3–77.5	Lymph node only (23.66%)	1. pembrolizumab (307)
						open-label	or metastatic			Visceral metastases (74.26%)	2. pembrolizumab + platinum-based CTX (351)
											3. platinum-based CTX (352)

## Data Availability

Data was extracted from published RCTs.

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
