# Peer review of "Immune Checkpoints Inhibitors and Chemotherapy as First-Line Treatment for Metastatic Urothelial Carcinoma: A Network Meta-Analysis of Randomized Phase III Clinical Trials"

_cancers, 2021, doi:10.3390/cancers13061484_

Round 1
Reviewer 1 Report
Chen HL, et al. present an original work on first line treatment for urothelial carcinoma. However, only 3 clinical trials are included, 2 of them are negative and the third one is pending for survival data. No data about the last updated results from the IMvigor 130 trials have been included. In addition, the combination of Chemotherapy plus immunotherapy is not approved by regulatory agencies, so the practical impact of the data presented are limited. The only consideration can be made for the immunotherapy monotherapy treatment. Introduction and Discussion should be improved focusing on the main issues and data in the first line setting, because most of the information is about second line therapies.Author Response
Thanks a lot for your suggestion. Although there were only 3 studies, all of them are high-quality randomized trials with more than 3000 patients in total. Per your suggestion, we have included the updated data of IMvigor 130 from ASCO-GU 2021, such as results from subgroup analysis. Baseline impact on OS was analyzed by age, gender, cisplatin eligibility and primary tumor site (Appendix D). Although combination of chemotherapy plus immunotherapy was not approved by regulatory agencies, potential benefit was observed from several emerging clinical trials. While the results of ICIs are very promising, how they compare with other ICIs is currently unknown. Hence, we performed this network meta-analysis on the available Phase III trials to identify the single best available treatment. We have modified our introduction and discussion focusing on the immunotherapy in the first line setting. Again, thank you for giving us the opportunity to strengthen our manuscript with your valuable comments.

Reviewer 2 Report
The paper by Cheng et al. presents a network meta analysis (NMA) of phase III studies on immune checkpoint inhibitors in combination with chemotherapy in the first-line treatment of metastatic urothelial carcinoma.
Of 1388 initial papers, 3 studies were ultimately included in the NMA after PRISMA workflow. These are the IMvigor130 study, the DANUBE study and the KEYNOTE 361 study.
OS, PFS, ORR and AEs were examined.
The authors conclude that, based on the available studies, it can be concluded that combination therapies can bring an advantage in the first-line therapy of mUC. In monotherapy, checkpoint inhibitors do not appear to be inferior to chemotherapy. No increased AEs appear to occur with combination therapy.
First of all, it is important to note that the question of this paper is highly relevant. mUCs are aggressive tumours that require optimal treatment. The clinical studies included are the ones relevant to this topic.
The authors describe their methodology in detail and comprehensibly. It would have been desirable to explain in more detail why an NMA was performed. Furthermore, there are no statements on whether the network was tested for conceptual heterogeneity or incoherence. The DANUBE study had a therapy arm with ICI+ICI, such an arm is missing in the KEYNOTE 361 and the IMvigor 130 study. Does this affect the analysis? Is the analysis valid at all? The authors need to comment on this.
It would also be desirable to briefly discuss the studies included. The chemotherapy arms (control arms) with 12.1 months (DANUBE), 13.1 months (IMvigor 130)
and 14.3 months (KEYNOTE-361) are roughly comparable. In contrast, the median follow-up was 8 months for KEYNOTE361 and 41.2 months for DANUBE. What are the possible implications of this?
The conclusion regarding OS is very vague. This could also be due to the fact that the analysis carried out does not clearly support the conclusion. All confidence intervals for the Hazard ratios for OS and some CIs for PFS, ORR and AEs include 1 and therefore provide insufficient evidence to conclude that the groups are statistically significantly different. The authors need to include this in their discussion.
It is absolutely clear that there is not much robust data available today for a meta-analysis on the issue under investigation. If these data cannot prove an advantage of a combination therapy, it is warranted to report this as a result. At least the results show a positive tendency, which gives hope.
An important result of the work is that monotherapy ICI is not inferior to chemotherapy. This is relevant in clinical practice, as it means that patients who are not suitable for chemotherapy can also receive therapy.
In summary, I consider the research question to be very relevant. The article certainly took a lot of effort and presents solid results in large parts. It would be desirable to improve the points mentioned, especially to clarify that the underlying method (NMA) was applied correctly.
Some minor issues:
On pages 2 and 3, it should read "PRISMA" diagram.
On page 4, Table 1, it should read "KEYNOTE 361".
On page 10, lines 226-228: "In our analysis, addition of pembrolizumab and atezolizumab to platinum-based chemotherapy resulted in better PFS and ORR when compared to CTX alone." - Using "and" is ambiguous.
In many instances, SCURA is used instead of SUCRA.
On page 11, line 277, it should read: "Unfortunately, we were unable to analyse this based on the existing data."
On page 11, line 305: Copenhagen is in Denmark.
On page 12, there are placeholder texts for Institutional Review Board Statement, Informed Consent Statement and Data Availability Statement
Author Response
Please find the attached file for your reference.
Reviewer (2)
From reviewer
The paper by Cheng et al. presents a network meta analysis (NMA) of phase III studies on immune checkpoint inhibitors in combination with chemotherapy in the first-line treatment of metastatic urothelial carcinoma. Of 1388 initial papers, 3 studies were ultimately included in the NMA after PRISMA workflow. These are the IMvigor130 study, the DANUBE study and the KEYNOTE 361 study. OS, PFS, ORR and AEs were examined. The authors conclude that, based on the available studies, it can be concluded that combination therapies can bring an advantage in the first-line therapy of mUC. In monotherapy, checkpoint inhibitors do not appear to be inferior to chemotherapy. No increased AEs appear to occur with combination therapy.
First of all, it is important to note that the question of this paper is highly relevant. mUCs are aggressive tumours that require optimal treatment. The clinical studies included are the ones relevant to this topic.
Reply
Thank you so much for your affirmation, I greatly appreciate you taking the time to write such a detailed response. Our replies are presented point by point below. Again, thank you for giving us the opportunity to strengthen our manuscript with your valuable comments.
Comment (1)
The authors describe their methodology in detail and comprehensibly. It would have been desirable to explain in more detail why an NMA was performed. Furthermore, there are no statements on whether the network was tested for conceptual heterogeneity or incoherence. The DANUBE study had a therapy arm with ICI+ICI, such an arm is missing in the KEYNOTE 361 and the IMvigor 130 study. Does this affect the analysis? Is the analysis valid at all? The authors need to comment on this.
Reply (1)
|
|
|
In the traditional meta-analysis, all included studies compare the same intervention with the same comparator. Unluckily, head to head comparisons are not always available. Network meta-analysis (NMA) is a technique for comparing interventions in a single analysis by combining direct and indirect evidence across the network of available studies. If two particular interventions have never been head to head compared, but these two have both been compared to a common comparator (placebo or standard therapy), then an indirect comparison can be evaluated versus the common comparator1-2. In our study (as shown in the figure below), direct evidence of ICI versus CTX, ICI+CTX versus CTX and ICI+ICI versus CTX were available. Indirect evidence was generated by using CTX as a common comparator for the comparison of all treatment arms in the network geometry. In addition, NMA allows ranking all treatment arms in the network geometry and identifying which is the best and worst based on the surface under cumulative ranking curve (SUCRA) As ICIs are very expensive and mUC patients have short-term survival, we urgently need to know which ICIs is most effective and reliable. NMA provides the evidences for clinicians and identify the single best available treatment in terms of efficacy and safety.
- Validation4
- Heterogeneity
Based on the figure above, only one RCT was included for all direct comparisons. For this reason, heterogeneity was not tested in our study.
- Consistency (or consistency)5
Inconsistency was defined as the difference between direct and indirect evidence. Design by treatment inconsistency model was used to check consistency for each direct treatment contrasts based on the difference between different study designs. Based on the figure above, there are 3 study designs (Design1: DANUBE , 3 arms with Durva+Treme v.s. Durva v.s CTX, Design2: KEYNOTE 361, 3 arms with Pembro v.s Prmbro+CTX v.s. CTX, Design3: IMvigor 130 , 3 arms with Atezo v.s Atezo+CTX v.s. CTX). No matter which treatment contrasts, only one design was presented (for example, treatment contrasts between Prmbro+CTX v.s. CTX, only design 2 was presented); therefore, no inconsistency was detected.
- Transitivity 4
Transitivity was evaluated by comparing the distribution of effect modifiers across different comparisons. Base on the table in the next page (Appendix E), balanced distribution was presented at baseline among patients received chemotherapy in DANUBE, KEYNOTE 361 and IMvigor 130. Therefore, chemotherapy was allowed to be the common comparator for a valid network comparison.
Reference
- Fernanda S. Tonin, Inajara Rotta, Antonio M. Mendes, Roberto Pontarolo. Network meta-analysis: a technique to gather evidence from direct and indirect comparisons. Pharm Pract (Granada). 2017 Jan-Mar; 15(1): 943.
- Diana M Sobieraj, Joseph C Cappelleri, William L Baker, Olivia J Phung, C Michael White, Craig Coleman. Methods used to conduct and report Bayesian mixed treatment comparisons published in the medical literature: a systematic review. BMJ Open. 2013; 3(7): e003111.
- Jeroen P Jansen1, Rachael Fleurence, Beth Devine, Robbin Itzler, Annabel Barrett, Neil Hawkins, Karen Lee, Cornelis Boersma, Lieven Annemans, Joseph C Cappelleri. Interpreting Indirect Treatment Comparisons and Network Meta-Analysis for Health-Care Decision Making: Report of the ISPOR Task Force on Indirect Treatment Comparisons Good Research Practices: Part 1Value Health. 2011 Jun;14(4):417-28
- Cochrane training handbook. Chapter 11: Undertaking network meta-analyses (https://training.cochrane.org/handbook/current/chapter-11)
- Dan Jackson, Jessica K Barrett, Stephen Rice, Ian R White, and Julian PT Higginsb. A design-by-treatment interaction model for network meta-analysis with random inconsistency effects .Stat Med. 2014 Sep 20; 33(21): 3639–3654.
Comment (2)
It would also be desirable to briefly discuss the studies included. The chemotherapy arms (control arms) with 12.1 months (DANUBE), 13.1 months (IMvigor 130) and 14.3 months (KEYNOTE-361) are roughly comparable. In contrast, the median follow-up was 8 months (median follow-up is 31.7 months instead of 8 months )for KEYNOTE361 and 41.2 months for DANUBE. What are the possible implications of this?
Reply (2)
Thank you for this important comment. Median follow up of included RCTs was extracted in the table below.
|
RCT |
Median follow up (month, IQR) |
|
IMvigor 130 |
11.8 (6.1,17.2) |
|
KEYNOTE-361 |
31.7 (22.0,42.3) |
|
DANUBE |
41.2 (37.9,43.2) |
- In RCT with shorter follow up duration, more subjects at
risk were being censored at cutoff point before death or disease progression. Therefore, different follow up duration may lead to bias.
- In order to decrease the potential bias, hazard ratio (HR) was used in our analysis instead of median months.
- Hazard ratio can be considered as an estimate of relative risk of interest over time. Due to the censored effects were presented in all interventions, bias from different follow up duration may be decreased among interventions with similar downward slope in survival curves.
- Figure above is the survival curve for OS in IMvigor 130. Orange line is median follow up point (11.8 months). Death events in blue or green zone were most likely to be censored if patients were included in the late stage of recruiting. In blue zone, similar downward slope was presented in both curves. Therefore, the censored effect was limited when calculating HR. In green zone, red line (chemotherapy) fallen with larger slope than blue line (Atezolizumab monotherapy). Therefore, the survival benefit was underestimated leaded to the worst SUCRA ranking was presented for patients with Atezolizumab( SCURA value for OS among Atezolizumab is 21.7%, for chemotherapy is 22.0%). We estimated the better OS performance will be shown for Atezolizumab users in IMvigor 130 with longer follow up duration in the future.
Comment (3)
The conclusion regarding OS is very vague. This could also be due to the fact that the analysis carried out does not clearly support the conclusion. All confidence intervals for the Hazard ratios for OS and some CIs for PFS, ORR and AEs include 1 and therefore provide insufficient evidence to conclude that the groups are statistically significantly different. The authors need to include this in their discussion.
It is absolutely clear that there is not much robust data available today for a meta-analysis on the issue under investigation. If these data cannot prove an advantage of a combination therapy, it is warranted to report this as a result. At least the results show a positive tendency, which gives hope.
An important result of the work is that monotherapy ICI is not inferior to chemotherapy. This is relevant in clinical practice, as it means that patients who are not suitable for chemotherapy can also receive therapy.
In summary, I consider the research question to be very relevant. The article certainly took a lot of effort and presents solid results in large parts. It would be desirable to improve the points mentioned, especially to clarify that the underlying method (NMA) was applied correctly.
Reply (3)
Thank you for the comments. Apart from P value and confidence interval (CI), one advantage of network meta analysis (NMA) is that NMA rank the treatments according to the likelihood. The surface under the cumulative ranking curve (SUCRA) is a numeric presentation of the overall ranking and presents a single number associated with each treatment. SUCRA values range from 0 to 100%. The larger SUCRA value was stood for the better rank of the intervention effects. Ranking based on SUCRA accounts better for the uncertainty in the estimated treatment effects. In our study, the probability of Atezo plus CTX was associated with the highest SUCRA for OS (SUCRA=80.2%), followed by Durva plus Treme (SUCRA=75.6%), Pembro plus CTX (SUCRA=70.6%), Pembro(SUCRA=50.8%), Durva (SUCRA=29.7%) , CTX alone (SUCRA=22.0%) and Atezo (SUCRA=21.7%). The SUCRA difference between ICI combination (more than 70%) and CTX alone (22%) is considerably large; it is positive information to extend the survival by ICI combination.
Reference : L Mbuagbaw, B Rochwerg, R Jaeschke, D Heels-Andsell, W Alhazzani, L Thabane , Gordon H Guyatt. Approaches to interpreting and choosing the best treatments in network meta-analyses. Syst Rev . 2017 Apr 12;6(1):79.
Comment (4)
Some minor issues:
- On pages 2 and 3, it should read "PRISMA" diagram.
- On page 4, Table 1, it should read "KEYNOTE 361".
- On page 10, lines 226-228: "In our analysis, addition of pembrolizumab and atezolizumab to platinum-based chemotherapy resulted in better PFS and ORR when compared to CTX alone." - Using "and" is ambiguous.
- In many instances, SCURA is used instead of SUCRA.
- On page 11, line 277, it should read: "Unfortunately, we were unable to analyse this based on the existing data."
- On page 11, line 305: Copenhagen is in Denmark.
- On page 12, there are placeholder texts for Institutional Review Board Statement, Informed Consent Statement and Data Availability Statement
Reply (4)
Thanks for your comments. All the mistakes were revised in the manuscript.

Round 2
Reviewer 1 Report
Authors have included the suggested recommendations.
I do not have further comments.
Thank you
Reviewer 2 Report
After the changes made, this paper presents relevant analysis to the topic.